# Outlining the Clinical Profile of *TCIRG1 14 Variants* including 5 Novels with Overview of ARO Phenotype and Ethnic Impact in 20 Egyptian Families

**DOI:** 10.3390/genes14040900

**Published:** 2023-04-12

**Authors:** Ghada Y. El-Kamah, Mennat I. Mehrez, Mohamed B. Taher, Hala T. El-Bassyouni, Khaled R. Gaber, Khalda S. Amr

**Affiliations:** 1Clinical Genetics Department, Human Genetics and Genome Research Institute, National Research Centre, Cairo 12622, Egypt; 2Oro-Dental Genetics Department, Human Genetics and Genome Research Institute, National Research Centre, Cairo 12622, Egypt; 3Prenatal Diagnosis and Fetal Medicine Department, Human Genetics and Genome Research Institute, National Research Centre, Cairo 12622, Egypt; 4Medical Molecular Genetics Department, Human Genetics and Genome Research Institute, National Research Centre, Cairo 12622, Egypt

**Keywords:** *TCIRG1* gene, malignant osteopetrosis, osteoclast, brittle bone, hypocalcification

## Abstract

*TCIRG1* gene mutations underlie osteopetrosis, a rare genetic disorder impacting osteoclast function with consequent brittle bones prone to fracture, in spite of being characterized by increased bone density. The disorder is known to exhibit marked genetic heterogeneity, has no treatment, and is lethal in most instances. There are reports of ethnic variations affecting bone mineral density and variants’ expression as diverse phenotypes even within individuals descending from the same pedigree. We herein focus on one of osteopetrosis’s three types: the autosomal recessive malignant form (MIM 259700) (ARO) that is almost always associated with severe clinical symptoms. We reviewed the results of about 1800 Egyptian exomes and we did not detect similar variants within our Egyptian dataset and secondary neurological deficit. We studied twenty Egyptian families: sixteen ARO patients, ten carrier parents with at least one ARO affected sib, and two fetuses. They were all subjected to thorough evaluation and *TCIRG1* gene sequencing. Our results of twenty-eight individuals descending from twenty Egyptian pedigrees with at least one ARO patient, expand the phenotype as well as genotype spectrum of recessive mutations in the *TCIRG1* gene by five novel pathogenic variants. Identifying *TCIRG1* gene mutations in Egyptian patients with ARO allowed the provision of proper genetic counseling, carrier detection, and prenatal diagnosis starting with two families included herein. It also could pave the way to modern genomic therapeutic approaches.

## 1. Introduction

Osteopetrosis is a rare genetic disease distinguished by flawed osteoclasts that has been linked to multiple genetic mutations with direct impact on osteoclast function with consequent large quantities of cortical bone deposition in a disorganized pattern, creating brittle bones prone to fracture [1].Defective osteoclast differentiation or function causes a disequilibrium of bone turnover, deformities, and impaired mineral homeostasis that gives rise to structural fragility [2]. Ethnic variations in bone mineral density (BMD) do not persistently follow special ethnic forms in fracture rates. Defining of fracture risk among populations should consider other factors besides BMD affecting bone strength such as bone structure and microarchitecture in addition to muscle mass, force generation, and anatomy as well as fat mass [3]. Coupled with the different ethnic rates, there are reports of the same variants expressed as diverse phenotypes ranging from no discernible phenotype to severe manifestations, even within individuals descending from the same pedigree [4].

Osteopetrosis is a heritable bone disorder that exhibits highly clinical and genetic heterogeneity, ranging in severity from benign to lethal in early childhood with overall disease incidence of about 1/20,000 [5,6]. Osteoporotic patients are characterized by reduction of marrow cavity, affecting hematologic function; related phenotypes are severe anemia, pancytopenia, frequent infections, and hepatosplenomegaly [7]. Three different forms of osteopetrosis have been described, based on the pattern of inheritance: autosomal recessive malignant form (MIM 259700) (ARO), autosomal dominant benign form (MIM 166600) (ADO), and X-linked mild as well as atypical forms that have also been reported (MIM 259710) [8]. Patients with ARO harbor mutations in different genes that are implicated in osteoclast function, almost always present with severe clinical symptoms in which the bone defect is associated with secondary neurological deficit and severe abnormalities including: macrocephaly, skull abnormality, hydrocephalus, progressive deafness, blindness, hepatosplenomegaly, severe anemia, and hypocalcemia [7]. 

ARO patients also suffer oral manifestations including delayed or complete failure of teeth eruption leading to impactions due to increased bone density [9]. Missing teeth have been reported as well [10]. The teeth may show crown and root malformations and enamel defects. Rampant caries are common and osteonecrosis, either spontaneous or following dental extractions, is a consistent finding owing to the decreased blood supply. It is worth noting that a dentist could be the first to diagnose these conditions [11,12]. Dentists must also be aware of osteopetrosis because of its effect on osteoclast function that causes impaired wound healing.

ARO, also known as infantile malignant osteopetrosis, has an incidence of 1:250,000 live births, with higher rates in specific geographic areas [13], and is caused by mutations in seven genes identified to date. The *TCIRG1* gene (MIM# 604592) [14,15], coding for a subunit of the osteoclast proton pump, is responsible for more than 50% of cases of human malignant osteopetrosis (ARO). The *TCIRG1* gene also has an alternative splicing and the usage of an ATG in exon 7, giving rise to another protein isoform TIRC7 which is a T cell-specific surface protein expressed upon lymphocyte activation of OC116 which plays a role in the interaction between the actin cytoskeleton and microtubules, and is essential for osteoclast ruffled border formation (Box 1) [16].

The six additional genes associated with human ARO are *TNFSF11, TNFRSF11A, CLCN7, OSTM1, SNX10,* and *PLEKHM1*. In addition, there is the carbonic anhydrase II gene (*CA2*; MIM# 259730), responsible for osteopetrosis with renal acidosis [17], the *CLCN7* gene (MIM# 602727), coding for a putative chloride channel, and the GL gene (*OSTM1*; MIM# 607649), the human homolog of the grey lethal (gL/gL) mouse mutant, that also account for a small proportion of ARO [18,19,20,21].

Our results of twenty-eight individuals descending from twenty Egyptian pedigrees with at least one ARO patient expand the phenotype as well as genotype spectrum of recessive mutations in the *TCIRG1* gene by five novel pathogenic variants. Identifying *TCIRG1* gene mutations in Egyptian patients with ARO allowed the provision of proper genetic counseling, carrier detection, and prenatal diagnosis starting with two families included herein. As a result, one healthy baby was born, and the second affected fetus was terminated. We hypothesized that the ARO patients from 20 Egyptian families may be associated with TCIRG1 gene mutations, thus whole exon sequencing of the TCIRG1 gene from the ARO patients was performed and linked to their clinical phenotypes.

## 2. Materials and Methods

The current study included 28 individuals: 16 ARO patients, 10 carrier parents with at least 1 affected ARO sib, and 2 fetuses through amniotic fluid sample analyses.

### 2.1. Clinical Evaluation

First, 26 individuals (16 patients and 10 carriers) belonging to 20 Egyptian families with at least 1 sib diagnosed as having autosomal recessive osteopetrosis (ARO), at the Hereditary Blood Disorders (HBD), Clinical Genetics, and Oro-Dental Genetics Clinics, Human Genetics and Genome Research Institute, National Research Center (NRC) Cairo, Egypt were registered.

Patients were diagnosed based on the presence of the cardinal phenotypic ARO disease manifestations following a detailed medical history recording including three-generation pedigree analyses, demographic data, history of present illness, and disease progression coupled with thorough clinical, hematological, radiological, and dental evaluation with emphasis on: growth, fractures, hematological status, evidence of cranial nerve compression, and neurological effects. Radiologic studies of bone and electroencephalography (EEG), and brain neuroimaging by computerized tomography (CT) and/or magnetic resonance, were recorded when available.

Five milliliters of blood were collected in EDTA tubes from all recruited patients and their available family members for genomic DNA extraction after obtaining informed written consent following thorough explanation and discussion according to the Helsinki Declaration of 1975, as revised in 1983.

### 2.2. Molecular Studies

DNA extraction using the QIA-gene extraction kit, according to the manufacturer’s protocol (Qiagen Inc., Valencia, CA, USA), was performed and polymerase chain reaction (PCR) standard amplification testing for all the encoding exons and exon–intron boundaries of the TCIRG1 gene was conducted using Taq DNA Polymerase (Qiagen, Inc.) and custom designed primer sets. Products of PCR amplification were purified with the QIAquick PCR Purification Kit (Qiagen, Inc.). The amplicons were sequenced in the forward and reverse directions using BigDye Terminator V3.1 Cycle Sequencing Kits (Applied Biosystems, Foster City, CA, USA) and an ABI PRISM^®^ 3500 capillary DNA analyzer (Applied Biosystems). Through the NCBI database, reference sequences of the *TCIRG1* gene (NG_007878.1), coding nucleotides (NM_006019.3), and amino acids (NP_006010.2) were retrieved.

Computational prediction of the effect of missense variants on VPP3 protein structure was carried out. The V-type proton ATPase 116 kDa subunit a3 (VPP3; Uniprot ID: Q13488), which is encoded by TCIRG1, has no entry in Protein Data Bank (PDB). The AlphaFold database provides predicted protein structure models covering the 830 amino acid residues of VPP3 [22]. PremPS was used to predict the impact of three missense variants: one that was newly detected in this study, p.R56W, p.L653R, and the recently reported p.Arg736Cys on the VPP3 structure, by (1) calculating the change in the unfolding free energy (∆∆G) or Gibbs free energy between folded and unfolded states of the protein, (2) showing the changes in noncovalent interactions between the mutated residues and their neighboring amino acid residues, and (3) information on the location of the mutated residue either on the surface or inside the protein’s core [23].

### 2.3. Prenatal Diagnosis

Prenatal diagnosis/counseling was provided for the mothers of two unrelated patients (sibs of ARO2 and 6). Amniocentesis was performed at 14 weeks of gestation and analyzed for the presence of the nonsense mutation and donor splice mutations described in their affected sibs, respectively.

## 3. Results

### 3.1. Phenotypic Results

Eight of the sixteen studied ARO patients were males with no gender preponderance. Patients’ ages ranged from 8 months to 17 years. All 16 cases presented with variable phenotypes; short stature was detected in 13 patients while height ranged within −0.4 to −1.8 SD for the rest of the cases and head circumference ranged within +1.8 to +3.7 SD. Three cases presented with microcephaly (−2.8 to −3.5 SD) however, frontal bossing was a cardinal sign (Figure 1). Fourteen patients had history of upper or lower limb fractures while one patient presented with several rib fractures, respiratory distress, and poor feeding. Anemia was a main complaint in 12/16 patients and abdominal U/S revealed hepatosplenomegaly in 75% of them. In patient ARO9, there was a right kidney lower polar multiloculated cystic lesion suggestive of polycystic kidney disease while the left kidney could not be visualized.

Clinical evaluation also revealed cardiac anomalies in two patients: large nonrestrictive patent ductus arteriosus, dilated left ventricle and left atrium, mild mitral regurge, and normal left ventricular systolic function in ARO9 and pericardial effusion in ARO3.

Neurological deficits were detected in 8/16 cases in the form of: developmental delay (25%), facial palsy (6.25%), deafness (12.5%), and blindness (6.25%) [Table 1]. No special phenotypes consistently corresponded to a certain variant or mutation type.

As for the oro-dental examination, four patients were examined only and the findings were as follows: premature loss of deciduous teeth, short philtrum (100%), delayed eruption (100%), thick alveolar ridge (75%), high-arched palate and enamel hypocalcification (75%), unusual pattern of eruption and asymmetry of the ridges and palate each in 50% of the cases, and abnormal occlusion and multiple lower frena and gingival recession in 25% of patients (Figure 2). On a different note, the pedigree analysis revealed that 90% of the probands were descended from consanguineous families and contained more than one affected sib (Figure 3).

### 3.2. Radiologic Results

Distinctive radiological findings of osteopetrosis were present in all 20 cases including increased density of long bones and pelvis with funnel-shaped (Erlenmeyer flask) deformity in some cases, straight mandibular angle, and acro-osteolysis of terminal phalanges (Figure 1B–F).

Brain magnetic resonance imaging detected brain atrophy in two cases (ARO1 and 7) (Figure 4).

### 3.3. Molecular Results

Sequence analysis of the *TCIRG1* gene in 28 individuals: 16 affected, 10 carriers, and 2 fetuses (through amniotic fluid samples analyses), identified 14 variants, including 5 novels of different mutation types. The characterized variants were inherited among our studied cohort in both monoallelic and/or biallelic patterns where a total of 40/42 *TCIRG1* gene alleles were identified.

Eleven unrelated patients harbored *TCIRG1* homozygous gene mutations while only three patients harbored compound heterozygous mutations and two patients had one null allele (GenBank^®^ record NM_006019.3). We could not detect a causative allele in two patients.

Among characterized variants, ten patients harbored five missense mutations, two novel and three previously reported, each has a single base substitution: four of which presented as homozygous mutations (p.Arg56Try, p.Gly458Ser, p.Leu653Arg, p.Pro775Arg) and the one missense mutation (p.Arg736Cys) represented a monoallelic mutation in a compound heterozygote patient. There was a predominance of the known missense mutation p.Pro775Arg in exon 19 which was detected in seven ARO patients (7/16, 44%) among our cohort.

Six splice mutation types, three novel and three reported, affecting eight families (8/20, 40%) are illustrated in Table 2. One reported base pair insertion of c.474_475insG (p.Pro161Alafs*66) was characterized in ARO13. Three patients (ARO1, 7, and 20) harbored compound heterozygous mutations. In patient ARO17, a novel small deletion of C at codon 465 c.1392_1392delC (p.Phe465Serfs*63) led to frameshift and protein termination after codon 62. Patient ARO6 harbored a reported nonsense mutation with substitution of C>T at nucleotide c.883C>T (p.Glu295*) that resulted in a truncated altered protein. In addition, analyses in four families (with history of affected/deceased ARO sib) seeking carrier detection were performed and parents were proved heterozygous for *TCIRG1* gene mutations that were always the same within each family. Accordingly, their sibs, who were not molecularly analyzed, are assumed as homozygous for the mutations (Table 2 and Figure 5).

All obtained results were matched with the published *TCIRG1* gene sequence (GeneBank^®^ EMBL No. NC_000011.10, NC_018922.2) using BLAST analysis of mutation sites and these variants are assumed to affect the protein since they were located in a conserved sequence among different vertebrate species (Table 2). Also, the pathogenicity of all new detected missense mutations were predicted using different Insilco analysis tools (Table 3).

### 3.4. Protein Prediction Results

The three missense mutations had positive ∆∆G values, which are evidence of protein destabilization. Leu653Arg and Arg736Cys had ∆∆G values of 0.86 and 0.82 kcal/mol, respectively, which are higher than the threshold of protein destabilization of 0.5 kcal/mol, unlike Arg56Trp which had a ∆∆G value of 0.45 kcal/mol. The three missense mutations affected α helices of the transmembrane of VPP3. Arg56Trp and Leu653Arg were predicted to be located on the protein surface while Arg736Cys was predicted to be inside the core of the protein. In Figure 6B, the consequence of Arg56Trp on noncovalent interactions is shown via the increased hydrophobic interactions between the aromatic residue of the mutated nonpolar tryptophan (Trp56) and the side chain of valine (Val52). The change from nonpolar leucine to the positively charged arginine (Leu653Arg) (Figure 6C) decreases hydrophobic bonding with leucine (Leu649) residue, and also introduces a van der Waals interaction with the carbonyl group of Leu649. In Figure 6D, the positively charged arginine wildtype residue is replaced by the mutated polar cysteine (Arg736Cys), therefore, the ionic interaction with negatively charged glutamic acid residue (Glu807) and polar interactions with histidine (His799) are lost. The aliphatic side chain of mutated cysteine is shorter than that of the arginine, therefore, weaker polar and van der Waals interactions with serine (Ser733) and tyrosine (Tyr734) were predicted (Figure 6).

### 3.5. Prenatal Diagnosis

The fetal DNA analysis for amniotic fluid sample No 1 (AF1/sib of ARO6) detected one heterozygous allele, c.883C>T, concluding the fetus was a carrier for the variant. A healthy newborn was delivered with no clinical, radiological, or laboratory abnormalities. The second (AF2/ARO2) fetal DNA sample No 2 revealed a homozygous splice site mutation, c.117+4A>T, concluding an affected fetus and the family decided on termination of pregnancy.

## 4. Discussion

Rare disorders such as ARO are not very rare in certain ethnic groups where consanguineous marriages are not unusual, among them are inhabitants of Africa and the Middle East where Egypt is situated at the crossroads between them. The rate of consanguineous marriage in Egypt is estimated to be about 33% [32], with consequent marked heterogeneous groups of genetic disorders.

A genotype-based understanding of rare diseases including data from population cohorts and clinical studies is crucial for understanding the causes of “phenotypic variabilities and expressivity of genetic variants in rare disease and across populations” [4]. In their study, Zengin et al., 2015, highlighted that “one size does not fit all”, urging the need to study variabilities in skeletal phenotype at different stages of life and in different populations both within and across continents [3].

In the current study, we discuss a cohort of twenty Egyptian families with at least one ARO affected member who presented to our clinics at the NRC exhibiting variable phenotypes as well as genotypes where fourteen variants were characterized with equally heterogeneous phenotypes compared to the ARO literature.

The diagnosis of osteopetrosis is usually based on clinical and radiological findings. The usual radiological feature is bone sclerosis and patients are usually macrocephalic. Three of our cohort (ARO7 and ARO8 and his affected sister) were microcephalic, they harbored compound heterozygous and homozygous missense mutations, respectively, but in different domains of the *TCIRG1* gene. Microcephaly is an atypical finding reported in a Turkish and two Indian cases’ reports [33,34,35]. Turkish and Egyptian people might share common ancestors because of the long Ottoman colonization of Egypt, and the other cases share the same ethnic (Indian) origin.

Reviewing the literature and, to our knowledge, there are no reports of ARO with normal stature, however, another unusual finding among our cohort is three patients (ARO11, 15, and 19) harboring different types of mutations, whose heights ranged within −0.4 to −1.8 SD according to Tanner staging. Li et al. suggested in their report in 2010 that, “although many human genes are found to have differential mRNA levels between populations, the extent of gene expression that could vary within and between populations largely remains elusive” [36]. Additionally, Kingdom and Wright, in 2022, suggested an overestimation of the penetrance and expressivity of rare disease variants compared to their effect on the general population and that they should be investigated across larger cohorts, possibly helping to reclassify variants previously considered completely penetrant.

Another atypical phenotype is the absence of any history of bone fractures in two patients (ARO1 and deceased sib of family 18) in spite of different ages at examination (8 months and 4 years) and different causative variants within different domains. Investigating muscle and bone relationships in ethnic groups was recommended by Zengin et al., to identify the effects of “differences in nutrition, cultural preferences, socioeconomic factors, sunshine exposure, and physical activity” on bone health within ethnic groups [3].

Oro-dental findings were mostly in concurrence with the literature except for the short philtrum, the high-arched palate, and the asymmetry of the ridges. The asymmetry of the ridges could be attributed to the defective osteoclastic activity and the uneven patterns of eruptions leading to this malformation. This study identified the enamel defects usually referred to in the literature. The defect was enamel hypocalcification and not hypoplasia. Defective mineralization was described by a study on mice where the authors concluded that enamel and dentine have defective mineralization [37].

In concordance with the phenotypic heterogeneity, studying the coding regions and exon–intron junctions of the TCIRG1 gene in our cohort of ARO patients and carriers disclosed fourteen variants including five novels which were not previously reported in the HGMD or Genome Aggregation Database (gnomAD). All published reports documented the genetic heterogeneity of this disease [6].

The new missense SNV, p.Arg56Try, and p.Leu653Arg caused alterations in the TCIRG1 transcript and a major shift based on amino acids’ molecular size and properties. Based on the clinical manifestations and pedigree analysis and according to the American College of Medical Genetics (ACMG) Variation Classification Guidelines, these mutations are classified as evidence for pathogenicity (PM) (Table 3) as they represent null mutations which might completely or partially abolish the function of the a3 subunit of the VPP.

The protein V-type proton ATPase subunit, isoform 3 (V-ATPase a3, VPP3, UniProtKB: Q13488), is translated by the TCIRG1 gene and consists of at least 13 different subunits, including the cytoplasmic V1 domain and the membrane-embedded V0 domain. The V1 domain is responsible for the hydrolysis of ATPase, and the V0 domain is responsible for a proton transporter [24,38]. The V0 domain is highly expressed in osteoclasts and can promote the function of osteoblasts to release protons and form an acidic environment, which is necessary for hydroxyapatite dissolution in the ruffled border of bone. The mutated TCIRG1 cannot be translated to form the correct V-ATPase complex on the osteoclast membrane [6].

Of note, identification of one null allele and probability in the presence of deep intronic mutation and/or the failure in detecting large allelic deletions could have led to the misinterpretation of the single variant sequence identified which could be erroneously considered as a homozygous mutation when masked by a coincident exonic deletion [30] Reviewing the literature, and according to the human gene mutation database, among a large portion of different TCIRG1 gene mutation types, the splice site mutation represents a total of 54/173 (HGMD; www.hgmd.org). Some of these splicing mutations affect the exon–intron junction or are located deeply within introns of the TCIRG1 gene. In the present study, eight ARO families harbored splicing defects that predominantly affected eleven alleles in ARO1, 2, 3, 5, 14, 15, 19, and 20 with five different splicing defects: three homozygous and four heterozygous splice mutations. Two of the five splice mutations were previously reported donor splice site mutations of intron 2 located in the N-terminus (c.117+1G>A and c.117+4A>T) in ARO1, 2, and 3 with abnormal processing of the transcripts. Functional analysis in some cases suggested that modified U1 snRNAs may represent a new therapeutic strategy for ARO patients with a U1 snRNP-dependent splicing defect [15,24,26,39] that is an important outcome of molecular analyses.

Among the six splice site mutations, five directly affect the donor or acceptor sites, while one homozygous one involves a more distant position, c.1021-9G>A, which might activate the cryptic splice site as in previous reports concerning the pathological role of different mutations far from the classical splice site in the TCIRG1 gene [24,27,39,40]. In ARO2, the homozygous splice donor site mutation (c.117+4A>T) was previously reported in a patient descending from a Turkish inbred family where functional analyses confirmed the presence of a cryptic splice site in the preceding exon resulting in a deletion of 14 amino acids (V26-D39del) of the N-terminal protein a3 subunit that had been proposed to interact with the V1 part of the proton pump complex involved in ATP binding and hydrolysis [15]. Substitutions in the splicing regulatory sequences are represented in a large portion, 40% (44 alleles), of the TCIRG1 variations in different ethnic groups. Herein, we report three new splice mutations in patients (ARO5, 15, and 20), c.713+2T>c, c.1021-9G>A, and c.2013+2T>G, located in the splice site of Int7, Int9, and Int16, respectively, and predicted to alter the splicing process, ultimately causing the disease phenotype. Similar to previous reports, the new variants might result in an altered protein through a not yet fully characterized effect.

Another disruption of protein due to the presence of a reported stop mutation was detected in ARO6, p.Gln295*, which falls within the sequence shared by TCIRG1 and TCIRG7 reported to cause a severe immunological phenotype in patients bearing this truncating variant. However, similar to our case, previously reported ARO cases were not necessarily associated with hypogammaglobulinemia. In addition, no genotype/phenotype correlations accompanied by the presence of defects in the T cell compartment have been reported [30].

Small deletions represent 18% (32/173) and small insertions 0.07% (12/173) of all TCIRG1 gene mutations reported in databases. We identified two frameshift mutations: a new deletion (c.1392_1392delC) in exon 12 and one reported insertion mutation, c.480dupG, in exon 5, each in one patient. The new frameshift mutation at codon 465 (c.1392_1392 delC) of ARO17 resulted in a truncated protein after 62 extraneous amino acids are translated from the wrong reading frame. Pangrazio et al., in 2012, reported another bp deletion, c.1393_1399del, within this hot spot in “codon 465”. The c.480dupG in exon 5 of ARO13 resulted in truncation of the protein product after 65 amino acids [30].

According to the HGMD, nearly 78% (108/173) of all reported mutations in the TCIRG1 gene may have been caused by a common NMD mRNA degradation mechanism. However, it is noteworthy that we could not detect these variants in more than 100 chromosomes, thus supporting the hypothesis of a pathologic role.

The impaired bone resorption associated with osteopetrosis-causing TCIRG1 mutations might result from defects in vesicle trafficking or fusion in osteoclasts as well as defects in the proton-pumping function of V-ATPase [13]. A total of five missense mutations, two new and three reported, were detected among our studied cohort. Although most of reported missense mutations among ARO families are not known to have common ancestors along the constructed pedigrees, we detected a common missense variant, p.Pro775Arg, located in the C-terminus of the V0 domain within seven families (31.5% (12/38 alleles)), suggesting Egyptian ARO patients might have a common ancestor. Consistent with our finding, two reported common missense variants (G405R and R444L) in evolutionarily conserved amino acid residues were inherited through families in a bi-allelic or mono-allelic state in pedigrees descending from a common ancestor [24]. However, there was no ascertainable difference in phenotypic presentation between our studied patients carrying the missense mutations from the other types of identified mutations leading to an alteration of the function of the a3 subunit of the VPP. Similarly, Susani et al., in 2004, detected some mutations as more frequent in a different ethnic population: c.1674–1G>A and c.2005C>T (p.Arg669X) were characterized in 17 and 16 alleles, respectively, constituting 30% of the TCIRG1 abnormalities among their studied patients originating from northern Europe and in West Flanders, Belgium, respectively [39].

Patients ARO7 and ARO20 harbored a newly detected missense variant located in the C-terminus, p.Arg736Cys, in a mono-allele where the arginine is changed to cysteine; this change is located at an evolutionarily highly conserved site. A missense mutation was reported in the same codon with a different base change, p.Arg736Ser, in a patient of European-American descent presenting with an abnormal hematological condition characterized by reduction in blood neutrophil count, which is inherited mostly as an autosomal dominant or, rarely, autosomal recessive pattern in different gene batteries [41]. Studies in mice have shown that although a homozygous mutation at amino acid 740 (homologous to human amino acid 736) is lethal in mice, a heterozygous mutation of the V-ATPase a3 subunit R740S causes dominant negative osteopetrosis [42].

Although we could not identify the second allele in the TCIRG1 gene in two patients (ARO14 and 19), the presence of only 50% gene dosage defect is not sufficient to cause disease, and we speculate different probabilities through complete gene analysis for mutations possibly located in the regulatory element or deep intronic region or these patients might have a large genomic deletion which could be undetected due to selective amplification of the other allele that could not be recognized through standard methods and/or completing the gene battery responsible for ARO disease [15].

Patient ARO9 who presented with a severe renal and cardiac phenotype harbored the same TCIRG1 c.2324C>G mutation as ARO8, 9, 10, 11, and 12, who did not suffer any cardiac or renal effects. In addition, some of the atypical phenotypes detected among our cohort were coinherited with some of the novel mutations characterized herein such as normal stature in ARO15 and 19 and no history of fracture in the deceased ARO18, however, the mutations were of different types in different domains of the TCIRG1 gene, making it difficult to conclude on a genotype–phenotype correlation.

ARO has no curative therapy, and genetic counseling is empirical for families having affected cases through either carrier detection and/or prenatal genetic testing. Among our cohort, four families, with deceased ARO sibs, seeking carrier detection were studied with confirmed presence of a null allele in both parents in each family. Additionally, prenatal diagnosis was carried out for two families and each mother received proper genetic counseling before and during their pregnancies.

## 5. Conclusions

In conclusion, our study contributes to the determination of ARO genetic heterogeneity both on the phenotypic and genotypic levels, with further outlining of more variants causing this devastating condition. Training of physicians and dentists on diagnosis of osteopetrosis is important to include them in proper genetic counseling/testing programs. Genetic testing is imperative in suspected cases of osteopetrosis as not many variants have yet been identified in children and early diagnosis will help carrier detection and disease prevention in addition to providing possible clues for mutation-based novel therapeutic approaches. Further investigation of the ethnic influence on gene expression in ARO phenotypes is recommended.

## Figures and Tables

**Figure 1 genes-14-00900-f001:**
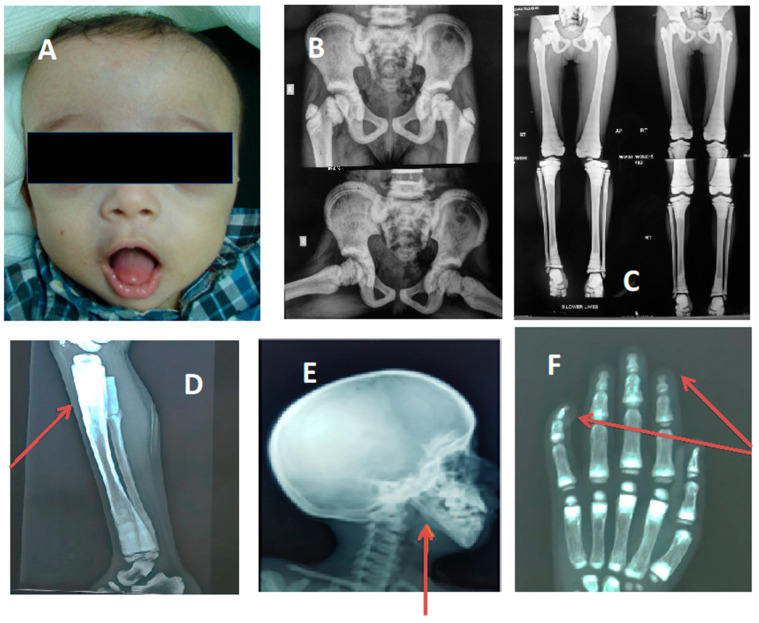
Some of cardinal criteria of ARO within our cohort; the red arrows point to some important radiological findings. (**A**) Frontal bossing; (**B**) increased bone density; (**C**,**D**) characteristic Erlenmeyer flask deformity; (**E**) straight mandibular angle; (**F**) bone-in-bone appearance and acro-osteolysis.

**Figure 2 genes-14-00900-f002:**
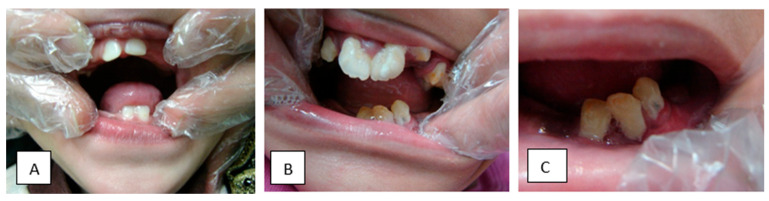
Some of the dental findings. (**A**) 3-year-old patient with delayed eruption of deciduous teeth; (**B**) 12-year-old patient with delayed eruption and hypocalcification; (**C**) Close-up of lower teeth showing recession of the gingiva.

**Figure 3 genes-14-00900-f003:**
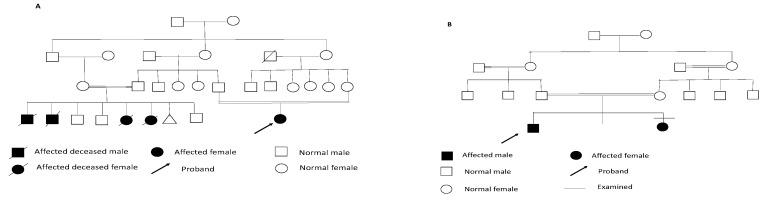
Examples of consanguineous pedigrees with more than one affected sib. (**A**) Pedigree of family of ARO5; (**B**) pedigree of family of ARO8.

**Figure 4 genes-14-00900-f004:**
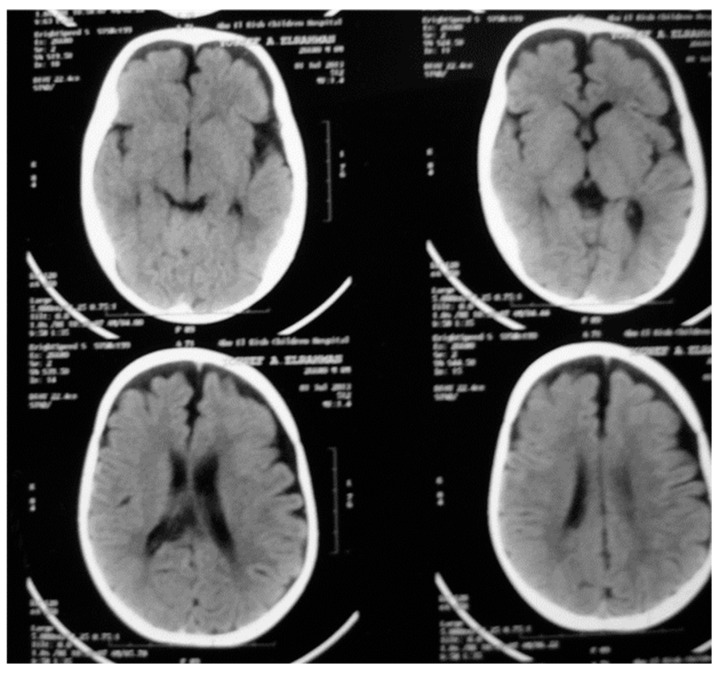
Brain magnetic resonance imaging of ARO1 showing atrophic changes.

**Figure 5 genes-14-00900-f005:**
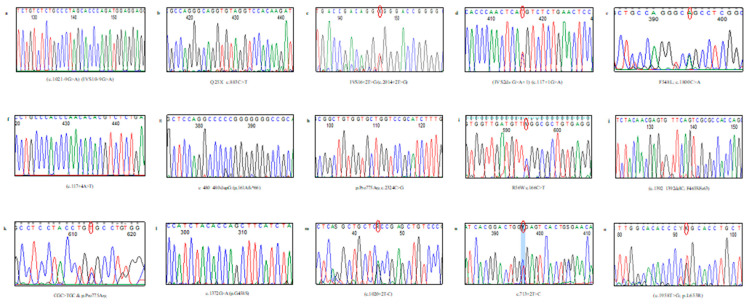
Different characterized *TCIRG1* gene variants.

**Figure 6 genes-14-00900-f006:**
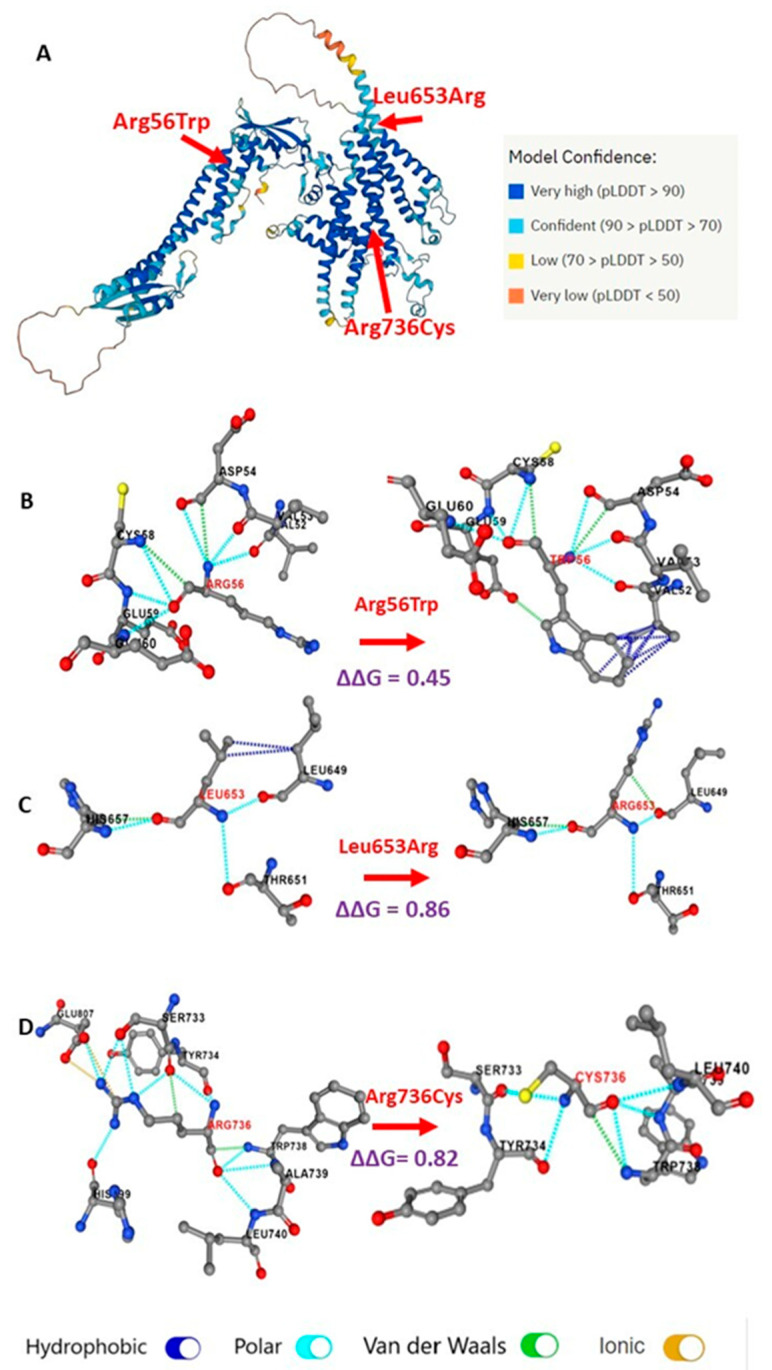
Predicted alterations of missense TCIRG1 variants (Arg56Trp, Leu653Arg, and Arg736Cys) on VPP3 protein structure. (**A**) Protein structure of V-type proton ATPase 116 kDa subunit a3 (VPP3; Uniprot ID: Q13488) predicted by AlphaFold. Per-residue confidence score (pLDDT) between 0 and 100 is shown in color codes. Red arrows show the position of the residues of interest: Arg56Trp and Leu653Arg on the protein surface, and Arg736Cys in the core of the protein. (**B**–**D**) The Prem PS software prediction of the changes in bonding between the mutated amino acid residues and the neighboring residues. The predicted changes in unfolding free energy (ΔΔG) values are calculated for each mutated protein. Noncovalent interactions are color coded at the bottom of the figure.

**Table 1 genes-14-00900-t001:** Summary of the clinical findings in the 16 ARO patients.

	Number of Patients	Percentage
Frontal bossing	16	100%
Short stature	13	81%
Macrocephaly	13	81%
Microcephaly	3	19%
History of fractures	15	94%
Anemia	12	75%
Hepato-splenomegaly	9	56%
Neurological deficit	8	50%
Cardiac effects	2	12.5%
Renal effects	1	6%

**Table 2 genes-14-00900-t002:** The characterized *TCIRG1* gene variants within the studied patients.

Patient Number	Nucleotide Change	Amino Acid Change	Location	Domain	Mutation Type	Status	References
ARO1	c.117+1G>A c.2324C>G	IVS2+1G>A p.Pro775Arg	Int2 Ex19	PD	Splice Missense	Heterozygous Heterozygous	[24,25]
ARO2	c.117+4A>T	p.V26_D39del	Int2	PD	Splice	Homozygous	[15]
ARO3	c.117+4A>T	IVS2+4A>T	Int2	PD	Splice	Homozygous	[15,26]
ARO4 # F. M.	c.166C>T c.166C>T	p.R56W p.R56W	Ex3 Ex3	CS	Missense Missense	Heterozygous Heterozygous Sib presumably homo	This study
ARO5	c.713+2T>C	IVS7+2T>C	Int7	-	Splice	Heterozygous Unch.	This study
ARO6	c.883C>T	Glu295 *	Ex9	CS	Stop codon	Homozygous	[27]
ARO7	c.2206C>T c.2324C>G	p.Arg736Cys p.Pro775Arg	Ex18 Ex19	TM7	Missense Missense	Heterozygous Heterozygous	[5,24,25,28]
ARO8a * ARO8b *	c.2324C>G c.2324C>G	p.Pro775Arg p.Pro775Arg	Ex19 Ex19	TM8	Missense Missense	Homozygous Homozygous	[24,25]
ARO9	c.2324C>G	p.Pro775Arg	Ex19	TM8	Missense	Homozygous	[24,25]
ARO10	c.2324C>G	p.Pro775Arg	Ex19	TM8	Missense	Homozygous	[24,25]
ARO11 # F. M.	c.2324C>G c.2324C>G	p.Pro775Arg p.Pro775Arg	Ex19 Ex19	TM8	Missense Missense	Heterozygous Heterozygous Sib presumably homo	[24,25]
ARO12 # F. M.	c.2324C>G c.2324C>G	p.Pro775Arg p.Pro775Arg	Ex19 Ex19	TM8	Missense Missense	Heterozygous Heterozygous Sib presumably homo	[24,25]
ARO13	c.480dupG	p.Pro161Alafs*66	Ex5	DD	Frameshift	Homozygous	[29]
ARO14 # F. M.	c.1020+2T-C c.1020+2T-C	IVS9+2T>C IVS9+2T>C	Int9 Int9	- -	Splice Splice	Heterozygous Heterozygous	[30]
ARO15	c.1021-9G>A	IVS10-9G>A	Int9	-	Splice	Homozygous	This Study
ARO16	c.1372G>A	p.Gly458Ser	Ex12	TM3	missense	Homozygous	[30,31]
ARO17	c.1392_1392delC	p.Phe465Serfs*63	Ex12	LL1	Frameshift	Homozygous	This study
ARO18 # F. M.	c.1958T>G c.1958T>G	p.L653R p.L653R	Ex16 Ex16	TM6	Missense Missense	Heterozygous Heterozygous Sib presumably homo	This study
ARO19	c.2013+2T>G	IVS16+2T>G	Int16	-	Splice	Heterozygous Unch.	This study
ARO20	c.2013+2T>G c.2206C>T	IVS16+2T>G p.R736C	Int16 Ex18	- TM7q	Splice missense	Heterozygous Heterozygous	This study [28]

* = affected sibs: ARO8a is a male sib, ARO8b his affected sister (counted two cases). # = carrier parents of a deceased affected sib. F. = father, M = mother, Ex = exon, Int = intron, Unch. = uncharacterized.

**Table 3 genes-14-00900-t003:** In silico analysis of novel missense mutations.

Mutation	PhD-SNP	PROVEAN	SNPs&GO	REVEL	SIFT	PolyPhen-2	Mutation Taster	MutPred	gnomAD Exome Frequencies
p.(R56W)	Disease (RI = 6)	Deleterious (−5.413)		Pathogenic (0.67)	Damaging (0)	Probably damaging (0.999)	Disease causing	Deleterious (0.591) Altered coiled coil (0.87; 0.0015) Loss of helix (0.28; 0.02)	ƒ = 0.0462
p.(L653R)	Disease (RI = 6)	Deleterious (−4.891)	Disease (RI = 2)	Pathogenic (0.8259)	Damaging (0.002)	Probably damaging (0.999)	Disease causing	Deleterious (0.581) Altered disordered interface (0.39; 0.0039) Gain of intrinsic disorder (0.35; 0.02)	Variant not found

PhD-SNP and SNPs&GO (predicting disease-associated variations from protein sequence and structure) directly predict disease effect and give reliability index range from 0 to 10; 0 unreliable and 10 reliable; predicting the functional effect of amino acid substitutions and indels (PROVEAN): score ≤ −2.5 is predicted as “Damaging”; otherwise it is predicted as “Neutral”; REVEL predicts pathogenicity by integrating scores from MutPred, FATHMM v2.3, VEST 3.0, PolyPhen-2, SIFT, PROVEAN, MutationAssessor, MutationTaster, LRT, GERP++, SiPhy, phyloP, and phastCons, PolyPhen-2 predicts possible impact of an amino acid substitution on the structure and function of a human protein, and MutPred integrates genetic and molecular data to reason probabilistically about the pathogenicity of amino acid substitutions. Score ranges from 0.0 (tolerated) to 1.0 (deleterious), MutPred also gives the predicted effect on the molecular mechanisms with *p*-values ≤ 0.05 (probability); SIFT predicts whether an amino acid substitution affects protein function, threshold for intolerance is 0.05; NE: not evaluated.

## Data Availability

Not applicable.

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
