# Peer review of "Outlining the Clinical Profile of TCIRG1 14 Variants including 5 Novels with Overview of ARO Phenotype and Ethnic Impact in 20 Egyptian Families"

_genes, 2023, doi:10.3390/genes14040900_

Round 1

Reviewer 1 Report

In this work El-Kamah et al. present a very novel work on the clinical profile of TCIRG1 14 in ARO phenotype. Obtaining such rare samples is very critical and praiseworthy . This work might be interesting to a specific group of researchers. I have two comments:

1. Is there any sex linked dependency of these mutations/novel variants ? Does ARO itself have any sex dependency?

2. Are these novel variants specific to Egyptian population? Is it possible to get similar datasets from public repositories and determine if these are present in global population?

Author Response

We would like to thank the reviewers for their efforts and valuable comments.

      1. Is there any sex linked dependency of these mutations/novel variants ? Does ARO itself have any sex dependency?

-No there is no sex dependency in ARO nor in  the novel variants

  1. Are these novel variants specific to Egyptian population? Is it possible to get similar datasets from public repositories and determine if these are present in global population?

-We reviewed the results of about 1800 Egyptian exomes and we did not detect similar variants within our Egyptian dataset however the novel variants were not previously reported within the international databases so we cannot actually confirm their ethnicity.

Reviewer 2 Report

The authors  herein focus on one of osteopetrosis’s three types; autosomal recessive ma-16 lignant form (MIM 259700) (ARO) that is almost always associated with severe clinical symptoms 17 and secondary neurological deficit. We studied 20 Egyptian families: 16 ARO patients, 10 carrier 18 parents with at least one ARO affected sib, and two fetuses . In conclusion, the study contributes to the determination of ARO genetic heterogeneity 441 both on the phenotypic and genotypic levels, with further outlining of more variants 442 causing this devastating condition. Training of physicians and dentists on diagnosis of 443 osteopetrosis is important to include them in proper genetic counselling/testing programs. Genetic testing is imperative in suspected cases of osteopetrosis as yet, not many  variants have been identified in children and early diagnosis will help carrier detection  and disease prevention in addition to providing possible clues for mutation based novel therapeutic approaches.  

The introduction is well written , with adequate bibliographic references . However a hypothesis of the study  must be included. Minor concern: Page 2, line 45 osteoporosis must be changed by osteopetrosis

The methodology is complete, widely described, which would allow the study to be carried out by another research group. Results : The phenotype and radiological data could be represented in the form of tables, which would facilitate their reading. The description of the genetic results is exhaustive, providing original data. The discussion is correct, adapting to the results obtained.  
However ,It's long and overly descriptive.

Author Response

We would like to thank the reviewers for their efforts and valuable comment.

  1. The introduction is well written, with adequate bibliographic references . However a hypothesis of the study  must be included.

Hypothesis was added (lines 95-99) and highlighted in yellow

  1. Minor concern: Page 2, line 45 osteoporosis must be changed by osteopetrosis

Could not find the osteoporosis in line 45 page 2 (it might have been corrected by the editor)

  1. The phenotype and radiological data could be represented in the form of tables, which would facilitate their reading.

Summary of the clinical findings in the 16 ARO patients was included in table 1 page 7

  1. The description of the genetic results is exhaustive, providing original data.The discussion is correct, adapting to the results obtained.  However ,It's long and overly descriptive.

The discussion was shortened as requested

Round 2

Reviewer 1 Report

Authors have satisfactorily responded to all comments.

Author Response

-I previously answered to all reviewers comments one by one
-I thanked the editor for his comments and included them